# Impact of SARS-CoV-2 on Ambient Air Quality in Northwest China (NWC)

**Shah Zaib** [1] , **Jianjiang Lu** [1,*], **Muhammad Zeeshaan Shahid** [2], **Sunny Ahmar** [3] **and Imran Shahid** [4]

1. Key Laboratory of Environmental Monitoring and Pollution Control of Xinjiang Bingtuan, School of Chemistry and Chemical Engineering, Shihezi University, Shihezi 832003, China; szaib716@gmail.com
2. Ministry of Public Health, Al Rumaila, Doha P.O. Box 42, Qatar; muhammad.shahid.1@kaust.edu.sa
3. Institute of Biological Sciences, University of Talca, 1 Poniente 1141, Talca 3460000, Chile; sunny.ahmar@yahoo.com
4. Environmental Science Center, Qatar University, Doha P.O. Box 2713, Qatar; ishahid@qu.edu.qa
* Correspondence: lujianjiang2015@163.com

**Abstract:** SARS-CoV-2 was discovered in Wuhan (Hubei) in late 2019 and covered the globe by March 2020. To prevent the spread of the SARS-CoV-2 outbreak, China imposed a countrywide lockdown that significantly improved the air quality. To investigate the collective effect of SARS-CoV-2 on air quality, we analyzed the ambient air quality in five provinces of northwest China (NWC): Shaanxi (SN), Xinjiang (XJ), Gansu (GS), Ningxia (NX) and Qinghai (QH), from January 2019 to December 2020. For this purpose, fine particulate matter ($PM_{2.5}$), coarse particulate matter ($PM_{10}$), sulfur dioxide ($SO_2$), nitrogen dioxide ($NO_2$), carbon monoxide (CO), and ozone ($O_3$) were obtained from the China National Environmental Monitoring Center (CNEMC). In 2020, $PM_{2.5}$, $PM_{10}$, $SO_2$, $NO_2$, CO, and $O_3$ improved by 2.72%, 5.31%, 7.93%, 8.40%, 8.47%, and 2.15%, respectively, as compared with 2019. The $PM_{2.5}$ failed to comply in SN and XJ; $PM_{10}$ failed to comply in SN, XJ, and NX with CAAQS Grade II standards (35 μg/m$^3$, 70 μg/m$^3$, annual mean). In a seasonal variation, all the pollutants experienced significant spatial and temporal distribution, e.g., highest in winter and lowest in summer, except $O_3$. Moreover, the average air quality index (AQI) improved by 4.70%, with the highest improvement in SN followed by QH, GS, XJ, and NX. AQI improved in all seasons; significant improvement occurred in winter (December to February) and spring (March to May) when lockdowns, industrial closure etc. were at their peak. The proportion of air quality Class I improved by 32.14%, and the number of days with $PM_{2.5}$, $SO_2$, and $NO_2$ as primary pollutants decreased while they increased for $PM_{10}$, CO, and $O_3$ in 2020. This study indicates a significant association between air quality improvement and the prevalence of SARS-CoV-2 in 2020.

**Keywords:** SARS-CoV-2; northwest China; AQI; primary pollutant; Pearson correlation



## 1. Introduction

In recent years, unprecedented industrial activity, urbanization, and motorization have jeopardized the air quality in northwest China (NWC) [1–4]. Multiple studies observed higher pollution levels in NWC due to increased industry, coal consumption, biomass burning, civil heating, power generation, urbanization, and natural sources [5–13]. Deteriorated air quality has attracted the attention of the scientific community because of its detrimental health effects [14–19]. Fine particulate matter ($PM_{2.5}$) is a major pollutant, ranked as the primary leading risk factor for disease in China, with more than 1.1 million premature deaths due to stroke, ischemic heart disease (IHD), chronic obstructive pulmonary disease (COPD), lung cancer, and lower respiratory infections [20–22].

The SARS-CoV-2 also known as "COVID-19" was first discovered in December 2019 in Wuhan, a city in the Hubei province of China [23–25]. This deadly virus covered the globe within two months, and the World Health Organization (WHO) declared a pandemic

on 11 March 2020 (WHO 2020). The common symptoms of COVID-19 include fever, dry cough, tiredness, and shortness of breath. In contrast, severe symptoms include a runny nose, sore throat, chills, muscle aches, headache, diarrhea, nausea, chest pain, breathing difficulties, and organ failure [26,27]. So far, COVID-19 has infected more than 133.10 million people and killed 2.90 million people worldwide (https://ourworldindata.org/covid-cases/, accessed on 14 April 2021).

To slow down the spread of the COVID-19 outbreak, China imposed a nationwide lockdown, e.g., travel restrictions, industrial closure etc., on 23 January 2020, starting from Wuhan (Hubei) [28]. Later on, the lockdown spread all over mainland China with a significant spatial and temporal variation. Multiple studies observed a reduction in anthropogenic emission and an improvement in the air quality index (AQI) due to epidemic prevention and control measures (lockdowns, travel restrictions, industrial closure etc.) throughout the globe [29–33]. Reference [34] observed that the average concentration of nitrogen dioxide ($NO_2$), $PM_{2.5}$, and coarse particulate matter ($PM_{10}$) decreased by 27.0%, 10.5%, and 21.4%, respectively, during lockdown (January to April) in China. Similarly, [35] observed short-term pollution reduction due to industrial closure, movement restrictions, and traffic stagnation in China. To further strengthen the claim that lockdowns improved air quality, [36] concluded that the AQI improved by 19.84 points in locked down cities and by 6.34 points in cities without formal lockdown. Reference [30] concluded that in Anhui province (Anqing, Hieifi, and Shuzou) the average concentrations of $PM_{2.5}$, $PM_{10}$, sulfur dioxide ($SO_2$), carbon monoxide ($CO$), and $NO_2$ decreased by 46.5%, 48.9%, 52.5%, 36.2%, and 52.8%, respectively, during the lockdown. Similarly, Hubei province (Wuhan, Jingmen, and Enshi) experienced 30.1%, 40.5%, 61.4%, 33.4%, and 27.9% reduction in $PM_{2.5}$, $PM_{10}$, $NO_2$, $SO_2$, and $CO$, respectively, during lockdown months (January 2020 to March 2020) [27]. Another study in Wuhan observed that the AQI, $PM_{2.5}$, and $NO_2$ decreased by 47.5%, 36.9%, and 53.3%, respectively, during lockdown [37]. Just like other areas, Shanghai also experienced a 9%, 77%, 31.3%, 60.4%, and 3% decrease in $PM_{2.5}$, $PM_{10}$, $SO_2$, $NO_2$, and $CO$, respectively, during lockdown months [38].

During the lockdown period (January 2020 to March 2020), most of the studies focused on air quality assessment in central China and nearby areas while ignoring rapidly developing areas of NWC [27,30,34–38]. However, very few studies assessed the air quality in northwest China (the industrial and manufacturing hub of China) to evaluate the influence of the COVID-19 outbreak and associated lockdowns on air quality. A study conducted by He et al. [31] observed that the AQI ($SO_2$, $PM_{2.5}$, $PM_{10}$, $NO_2$, and $CO$) improved by 7.8% (6.76%, 5.93%, 13.66%, 24.67%, and 4.58%) in Lanzhou (Gansu province) during the lockdown. Similarly, [39] observed that $PM_1$ decreased by 50% during the Lanzhou lockdown period (January to March). Depending on the spread of the viral outbreak, lockdowns, travel restrictions, etc. were extended and re-imposed in some areas, e.g., Kashgar city (October 2020). No study has extensively analyzed the influence of the viral outbreak and lockdowns on air quality in northwest China.

In this study, we assessed the spatial and temporal variation of air pollution in 53 cities of NWC. We examined six criteria pollutants ($PM_{2.5}$, $PM_{10}$, $SO_2$, $NO_2$, $CO$, and $O_3$), the air quality index (AQI), the proportion of AQI classes, and major pollutants etc. for a period of two years (January 2019 to December 2020) to illustrate the impact of SARS-CoV-2 and associated lockdowns on the spatial and temporal distribution of air pollution in NWC. We believe this is the first study focusing on the long-term impact of the COVID-19 outbreak on air quality in 53 cities of NWC and is of considerable significance to environmental protection and human health.

## 2. Materials and Methods

### 2.1. Site Selection

Northwest China (NWC) is a mixture of agricultural land, deserts, mountains, industrial complexes, significant mineral reserves, and degraded air quality. We analyzed the ambient air quality in 53 cities locatecd in five provinces (Shaanxi (SN), Xinjiang (XJ),

Gansu (GS), Ningxia (NX), and Qinghai (QH)) of NWC from January 2019 to December 2020 to understand the spatial temporal variation across NWC better. SN includes 10 cities (Ankang, Baoji, Hanzhong, Shanglou, Tongcuan, Weinan, Xian, Xianyang, Yanna, Yulin), and XJ includes 16 cities (Aksu, Altay, Bortala, Crete, Changji, Hami, Hotan, Ili, Karamay, Korla, Kashgar, Shihezi, Tacheng, Turpan, Urumqi, Wujiaqu). Similarly, GS includes 14 cities (Dingxi, Gannn, Jiayuguan, Jinchang, Jiuquan, Lanzhou, Linxia, Longnan, Pinglian, Qingyang, Silver city, Tianshui, Wuwei, Zhangye), and NX includes five cities (Guyuan, Shizuishan, Yinchuan, Wuzhong, Zhongwei). QH includes eight cities (Guolou, Haibei, Haidong, Hainan, Haixi, Huangnan, Xinning, Yushu). A location map of the cities under observation is shown in Figure 1.

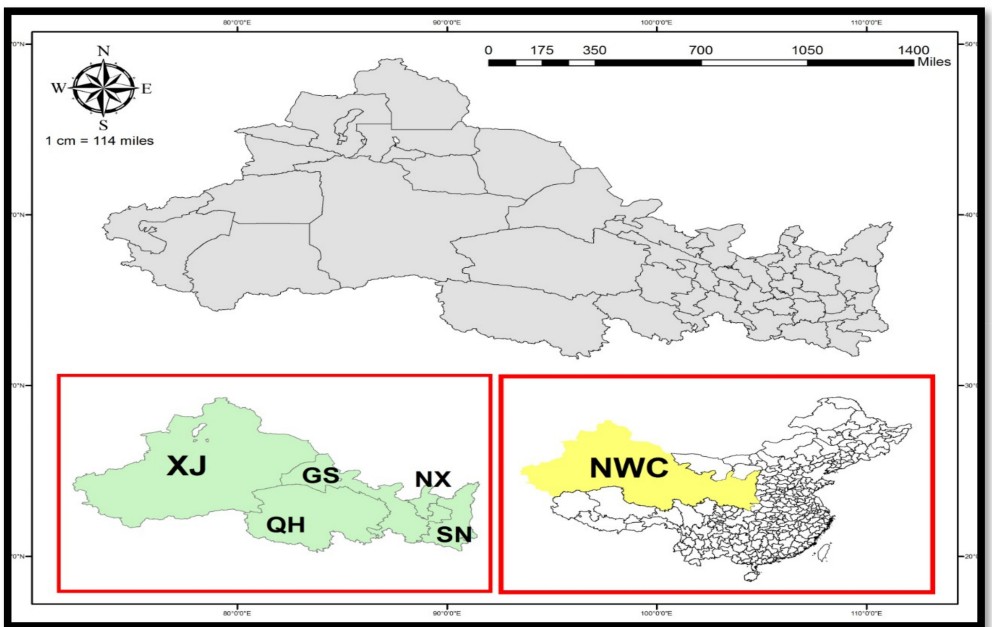

**Figure 1.** Location map of five provinces (Shaanxi (SN), Xinjiang (XJ), Gansu (GS), Ningxia (NX), Qinghai (QH)) in Northwest China (NWC).

### 2.2. Data Collection

The hourly concentration of six criteria pollutants ($PM_{2.5}$, $PM_{10}$, $SO_2$, $NO_2$, CO, and $O_3$) in NWC was obtained from the China National Environmental Monitoring Center (CNEMC) for a period of two years (2019–2020). The online data-sharing platform had installed hundreds of ambient air monitoring stations, covering both urban and rural areas in 337 cities of China, and published information according to the Technical Guideline on Environmental Monitoring Quality Management (HJ 630-2011).

### 2.3. Air Quality Index (AQI)

The air quality index (AQI) is a color-coded scale that simplifies different pollutants' concentrations into a single numerical value to reflect overall air quality, health effects, sensitive groups, and required precautionary measures. The AQI includes 24-h average measurements of $NO_2$, $SO_2$, CO, PM2.5, PM10, and 8-h average concentrations of $O_3$ [12,13,40]. When the AQI is higher than 50, the highest Individual air quality index (IAQI) is considered a primary pollutant for that given day [13,30,41–43]. The individual air quality index (IAQI) for six criteria pollutants is determined by using Equation (1), and the overall AQI is calculated based on the highest IAQI by using Equation (2) according to the instruction given in the technical regulation on ambient air quality index (on trial) (HJ-633-2012).

$$IAQI_P = I_{high} - I_{low} / C_{high} - C_{low} * (C_P - C_{low}) + I_{low} \tag{1}$$

$IAQI_P$ = Individual sub air quality index of the pollutant p
$C_P$ = concentration of the pollutant p
$C_{high}$ = concentration breakpoint that is $\geq C_p$
$C_{low}$ = concentration breakpoint that is $\leq C_p$
$I_{high}$ = index breakpoint corresponding to $C_{high}$
$I_{low}$ = index breakpoint corresponding to $C_{low}$

$$AQI = \max (I_1, I_2, \ldots , In) \tag{2}$$

*AQI* has the following six categories: Class I; 0–50 (green), good; Class II: 51–100 (yellow), moderate; Class III: 101–150 (orange), unhealthy for sensitive groups; Class IV: 151–200 (red), unhealthy; Class V: 201–300 (purple), very unhealthy; Class VI: 300–500 (maroon), hazardous.

### 2.4. Quality Assurance and Quality Control (QA&AR)

Quality assurance and control procedures for ambient air quality data were strictly by Chinese Ambient Air Quality Standards (CAAQS) (GB 3095 2012). The daily average value was calculated when we had valid data for more than 16 h of that day (except for $O_3$, minimum 6-h values for 8-h $O_3$ value). The monthly average was calculated when we had 27 daily mean values; an annual value was calculated when we had 324 daily mean values. Besides this, manual inspection was carried out to remove abnormal values e.g., $PM_{2.5}$ values higher than $PM_{10}$ values.

### 2.5. Kriging (Ordinary/Universal)

Kriging is a geospatial interpolation technique that defines the unknown values depending on the available known values and considers both the distance and the degree of variation between known data points when estimating values in unknown areas. We used kriging (ordinary) to evaluate the spatial distribution of criteria pollutants ($PM_{2.5}$, $PM_{10}$, $SO_2$, $NO_2$, CO, $O_3$), AQI, etc., in NWC, and then applied reclassification to obtain the desired map format.

### 2.6. Statistical Analysis

In this study, we used the Statistical Package for Social Sciences (SPSS) for Windows (IBM SPSS Statistics, Version 25) to evaluate Pearson's correlation coefficient for criteria pollutants on an annual and seasonal basis. Pearson's correlation is a correlation coefficient commonly used in linear regression and used to measure the strength of relationships between the six air pollutants. The effect of a certain variable was considered statistically significant for P (0.01 and 0.05) (two tailed). Annual mean values of data were used for the analysis of six criteria pollutants between 2019 to 2020: mean absolute deviation (MAD), mean square error (MSE), root mean square error (RMSE), mean absolute percentage error (MAPE), and mean percentage error (MPE), and were calculated by Excel 2016.

### 3. Results

#### 3.1. Annual and Seasonal Changes in Criteria Pollutants

During the study period (2019–2020), the annual average concentration of $PM_{2.5}$, $PM_{10}$, $SO_2$, $NO_2$, CO, and $O_3$ (Table S1) improved by 2.72%, 5.31%, 7.93%, 8.40%, 8.47%, and 2.15%, respectively, in NWC (Figure 2). The annual average concentration of $PM_{2.5}$ exceeded CAAQS Grade II standards (35 µg/m$^3$, annual mean) in SN (24.91%), XJ (38.94%), and NWC (6.1%) (Figure 2a). Similarly, the annual average concentration of $PM_{10}$ exceeded CAAQS Grade II standards (70 µg/m$^3$, annual mean) in SN (8.71%), XJ (72.13%), NX (12.45%), and NWC (18.74%) (Figure 2b). While $SO_2$ and $NO_2$ complied with CAAQS Grade II standards (20 µg/m$^3$ and 40 µg/m$^3$, annual mean) in NWC (Figure 2c,d). CO and $O_3$ do not have annual standards under CAAQS; both the CO and $O_3$ decreased in SN, XJ, GS, NX, QH, and NWC during 2020. The highest concentration of $PM_{2.5}$, $PM_{10}$,

$SO_2$, $NO_2$, CO, and $O_3$ occurred in XJ, XJ, NX, SN, XJ, and QH, respectively. Figure 3 explains the spatial distribution of criteria pollutants in 53 cities of NWC during 2019 and 2020, obtained by kriging (ordinary) interpolation technique. The obtained results from spatial interpolation (kriging) were quite similar to the actual values. In 2020, 34 (64.15%), 39 (73.58%), 38 (71.7%), 47 (88.68%), 44 (83.01%), and 38 (71.7%) cities of NWC experienced a reduction in $PM_{2.5}$, $PM_{10}$, $SO_2$, $NO_2$, CO, and $O_3$, respectively (Figure 3).

PM2.5, $SO_2$, $NO_2$, and CO observed the highest value in winter and lowest in summer in a seasonal variation. The concentration of $PM_{10}$ was highest in spring, while $O_3$ was highest in summer and lowest in winter. The average concentration of $PM_{2.5}$, $PM_{10}$, $SO_2$, $NO_2$, and CO decreased in winter, spring, and summer 2020. In autumn 2020, $PM_{2.5}$, $PM_{10}$, $SO_2$, and $NO_2$ increased while CO decreased. $O_3$ decreased in spring, summer, and autumn while it increased in winter 2020 as compared with 2019 (Figure 4). $PM_{2.5}$ exceeded the daily limits of CAAQS Grade II (75 $\mu g/m^3$) during winter in SN (2019), and XJ (2019, 2020) (Figure 4a), while $PM_{10}$ exceeded the daily limits of CAAQS Grade II (150 $\mu g/m^3$) during spring in XJ (2019, 2020) (Figure 4b). Gaseous pollutants ($SO_2$, $NO_2$, CO, and $O_3$) complied with daily limits of CAAQS Grade II (150 $\mu g/m^3$, 80 $\mu g/m^3$, 4 $mg/m^3$, and 160 $\mu g/m^3$) in SN, XJ, GS, NX, QH, and NWC (Figure 4c–f).

### 3.2. $PM_{2.5}/PM_{10}$ Ratio

During the study period (2019–2020), the average $PM_{2.5}/PM_{10}$ ratio in NWC ranged from 0.325 ± 0.135 to 0.640 ± 0.190 with an average of 0.472 ± 0.100, highest $PM_{2.5}/PM_{10}$ ratio occurring in SN followed by QH, NX, GS, and XJ, and experiencing an average change of 2.56% in NWC in 2020 as compared with 2019 (Figure 2h). In a seasonal variation, the highest $PM_{2.5}/PM_{10}$ ratio occurred in winter followed by autumn, summer, and spring, respectively, and increased by 10.68% in winter in NWC during 2020 (Figure 4h). In 2020, 62.26% of cities observed an increase in the $PM_{2.5}/PM_{10}$ ratio. Similarly, spring, summer, autumn and winter experienced an improvement in 33.96%, 52.83%, 54.72%, and 90.57% cities of NWC, respectively, in 2020 against 2019 (Figure 5).

### 3.3. Air Quality Index (AQI)

During the study period (2019–2020), the average AQI in NWC ranged from 43.34 ± 10.15 to 194 ± 210.26 with an average of 79.65 ± 19.90, and the highest AQI occurred in XJ followed by SN, GS, NX, and QH. The AQI improved by 4.67% (10.26%, 2.25%, 2.73%, 0.31%, 9.74%) in NWC (SN, XJ, GS, NX, QH) during 2020 as compared with 2019 (Figure 2g). In a seasonal variation, the highest AQI occurred in winter followed by spring, summer, and autumn, respectively, and improved by 6.69%, 2.91%, 8.57%, and 1.59%, respectively, in NWC during 2020 (Figure 4g). Seasonal variation was not consistent throughout NWC e.g., SN and XJ experienced the highest AQI in winter, GS and NX in spring. At the same time, QH observed the highest AQI in summer. Figure 6 illustrates the annual and seasonal spatial distribution of the AQI in NWC. In 2020, 77.36% cities experienced an improvement in the AQI. Similarly, spring, summer, autumn and winter experienced an improvement of 83.02%, 13.21%, 52.83%, and 62.26% in the cities of NWC, respectively, in 2020 against 2019. Significant improvement in the AQI occurred in winter (December to February) and spring (March to April) when the viral outbreak, lockdown, and movement restrictions were at their peak.

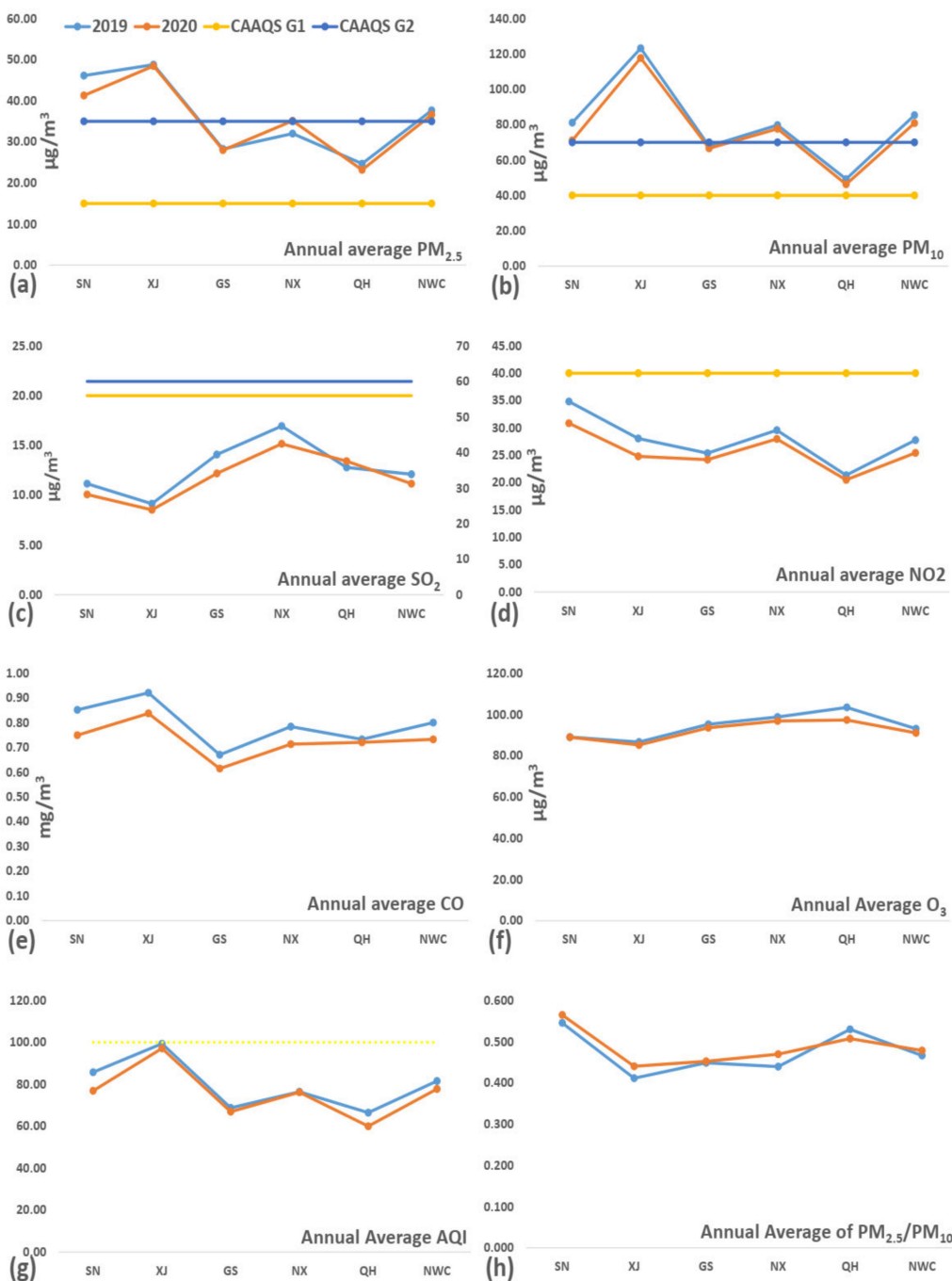

**Figure 2.** Influence of SARS-CoV-2 on the annual variation of six criteria pollutants: PM$_{2.5}$ (**a**), PM$_{10}$ (**b**), SO$_2$ (**c**), NO$_2$ (**d**), CO (**e**), and O$_3$ (**f**)), AQI (**g**), and PM2.5/PM$_{10}$ (**h**) in five provinces (Shaanxi (SN), Xinjiang (XJ), Gansu (GS), Ningxia (NX), and Qinghai (QH)) of northwest China (NWC) during 2019–2020. Descriptions are as follows: blue line (2019), orange line (2020), yellow line (CAAQS Grade I), blue line (CAAQS Grade II), and yellow dotted line (AQI threshold). The abbreviations are as follows: PM$_{2.5}$ (fine particulate matter), PM$_{10}$ (coarse particulate matter), SO$_2$ (Sulfur dioxide), NO$_2$ (nitrogen dioxide), CO (carbon monoxide), O$_3$ (ozone), CAAAQS (Chinese Ambient Air Quality Standards).

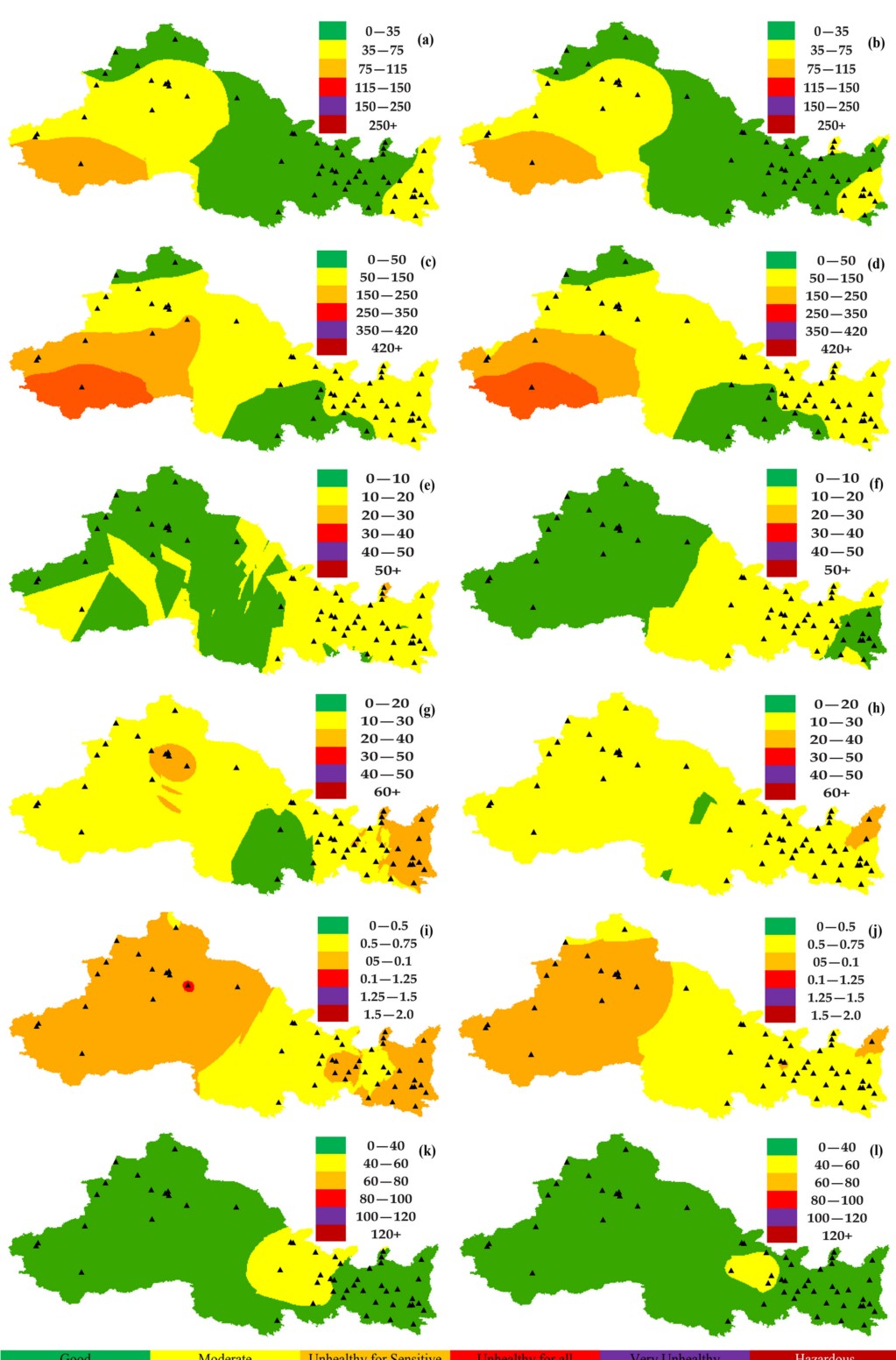

**Figure 3.** Spatial distribution of PM$_{2.5}$ (**a,b**), PM$_{10}$ (**c,d**), SO$_2$ (**e,f**), NO$_2$ (**g,h**), CO (**i,j**), and O$_3$ (**k,i**) in northwest China (NWC) during 2019 and 2020. Colors represent the different pollution levels e.g., green (good), yellow (moderate), orange (unhealthy for the sensitive group), red (unhealthy for all), purple (very unhealthy), and maroon (hazardous). The abbreviations are as follows: PM$_{2.5}$ (fine particulate matter), PM$_{10}$ (coarse particulate matter), SO$_{2.}$ (Sulfur dioxide), NO$_2$ (nitrogen dioxide), CO (carbon monoxide), and O$_3$ (ozone).

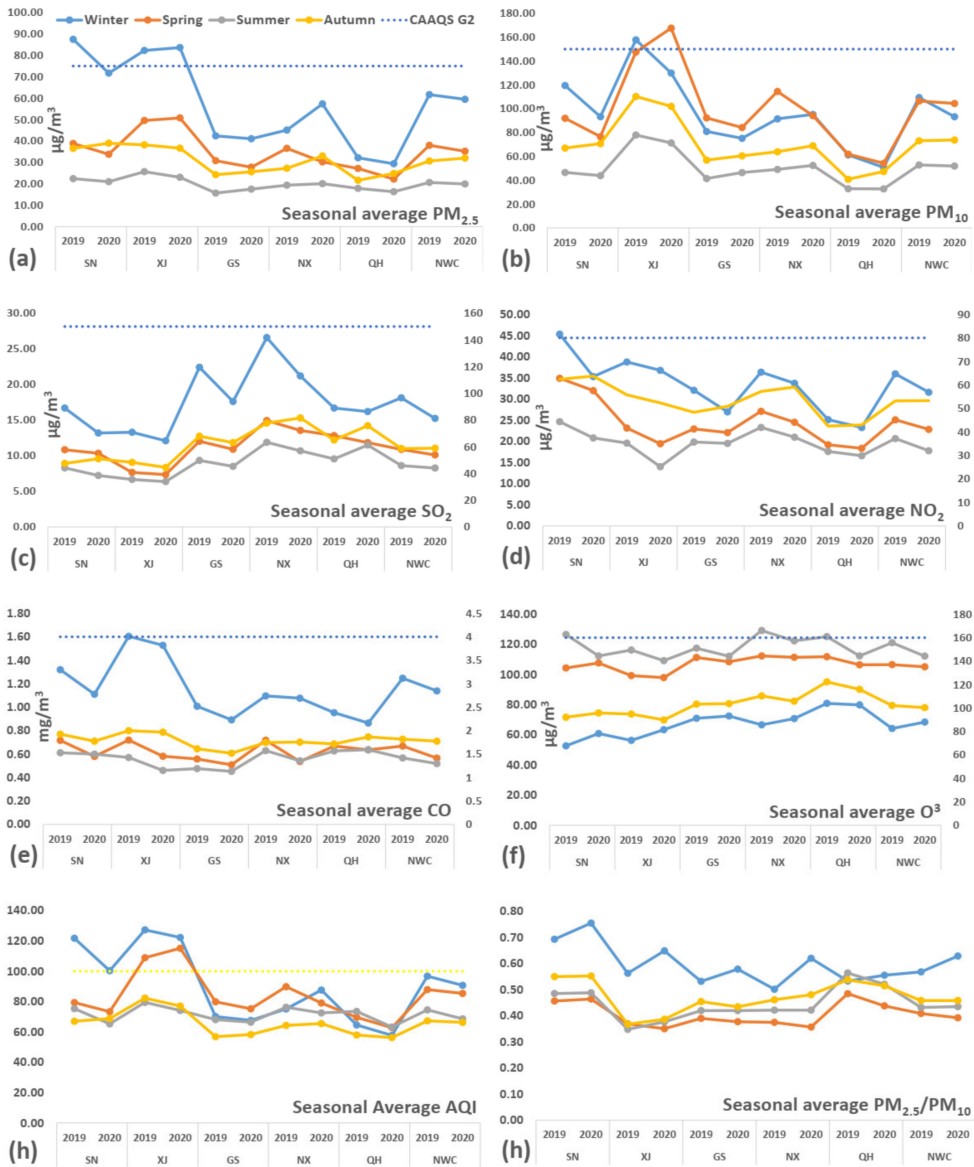

**Figure 4.** Influence of SARS-CoV-2 on seasonal variation of six pollutants: PM$_{2.5}$ (**a**), PM$_{10}$ (**b**), SO$_2$ (**c**), NO$_2$ (**d**), CO (**e**), O$_3$ (**f**); AQI (**g**), and PM$_{2.5}$/PM$_{10}$ ratio (**h**) in five provinces (Shaanxi (SN), Xinjiang (XJ), Gansu (GS), Ningxia (NX), and Qinghai (QH)) of northwest China (NWC) during 2019–2020. Descriptions are as follows: blue line (winter), orange line (spring), gray line (summer), yellow line (autumn), blue dotted line (CAAQS Grade II), and yellow dotted line (AQI threshold). The abbreviations are as follows: PM$_{2.5}$ (fine particulate matter), PM$_{10}$ (coarse particulate matter), SO$_2$ (Sulfur dioxide), NO$_2$ (nitrogen dioxide), CO (carbon monoxide), O$_3$ (ozone), PM$_{2.5}$/PM$_{10}$ (ratio of PM$_{2.5}$ with PM$_{10}$), and AQI (air quality index).

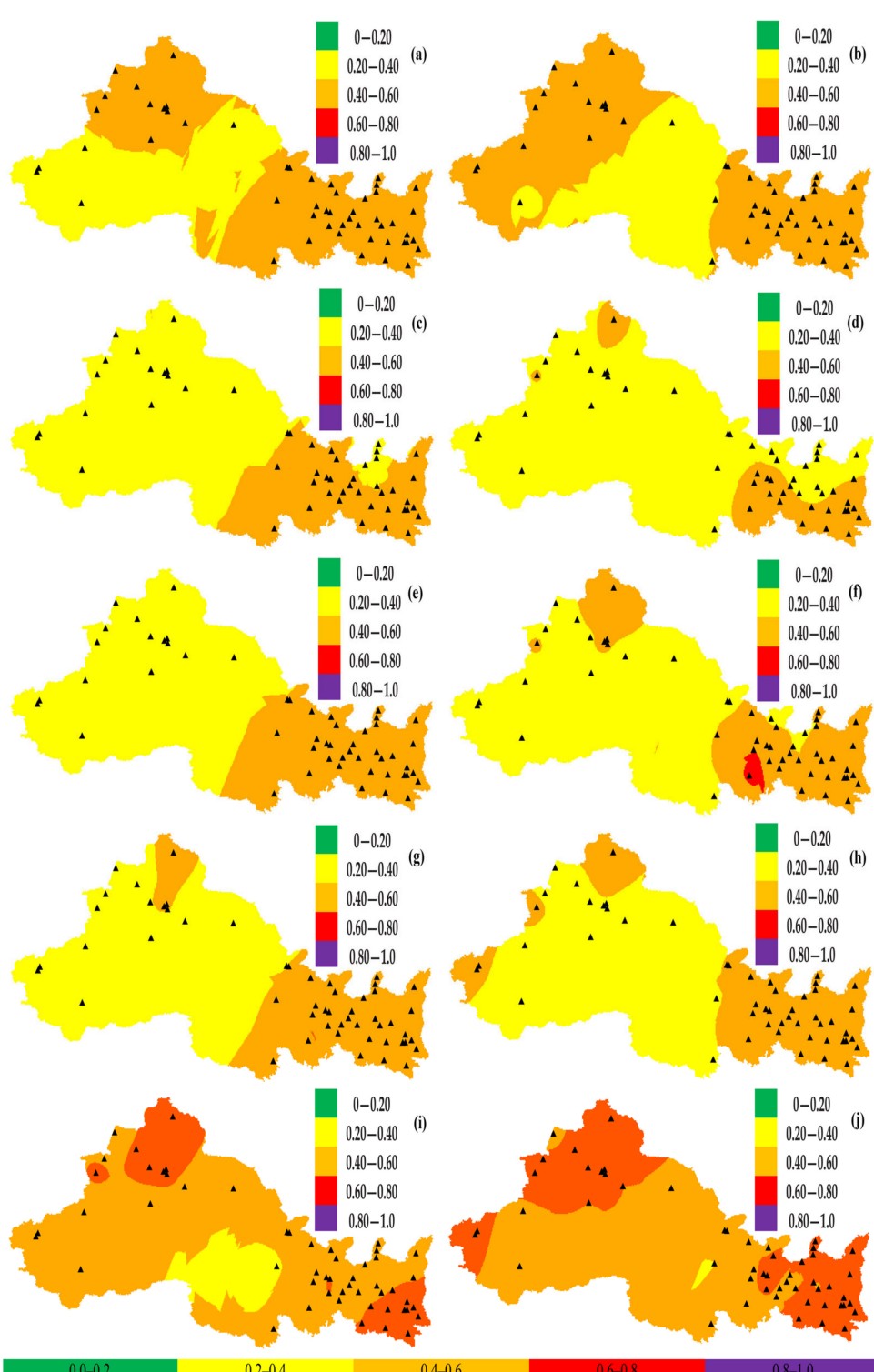

**Figure 5.** Annual (**a**,**b**) and seasonal (spring (**c**,**d**), summer (**e**,**f**), autumn (**g**,**h**), winter (**i**,**j**)) variation of $PM_{2.5}/PM_{10}$ ratio in northwest China (NWC) during 2019 and 2020. Colors represent the different pollution levels e.g., green (good), yellow (moderate), orange (unhealthy for the sensitive group), red (unhealthy for all), and purple (very unhealthy).

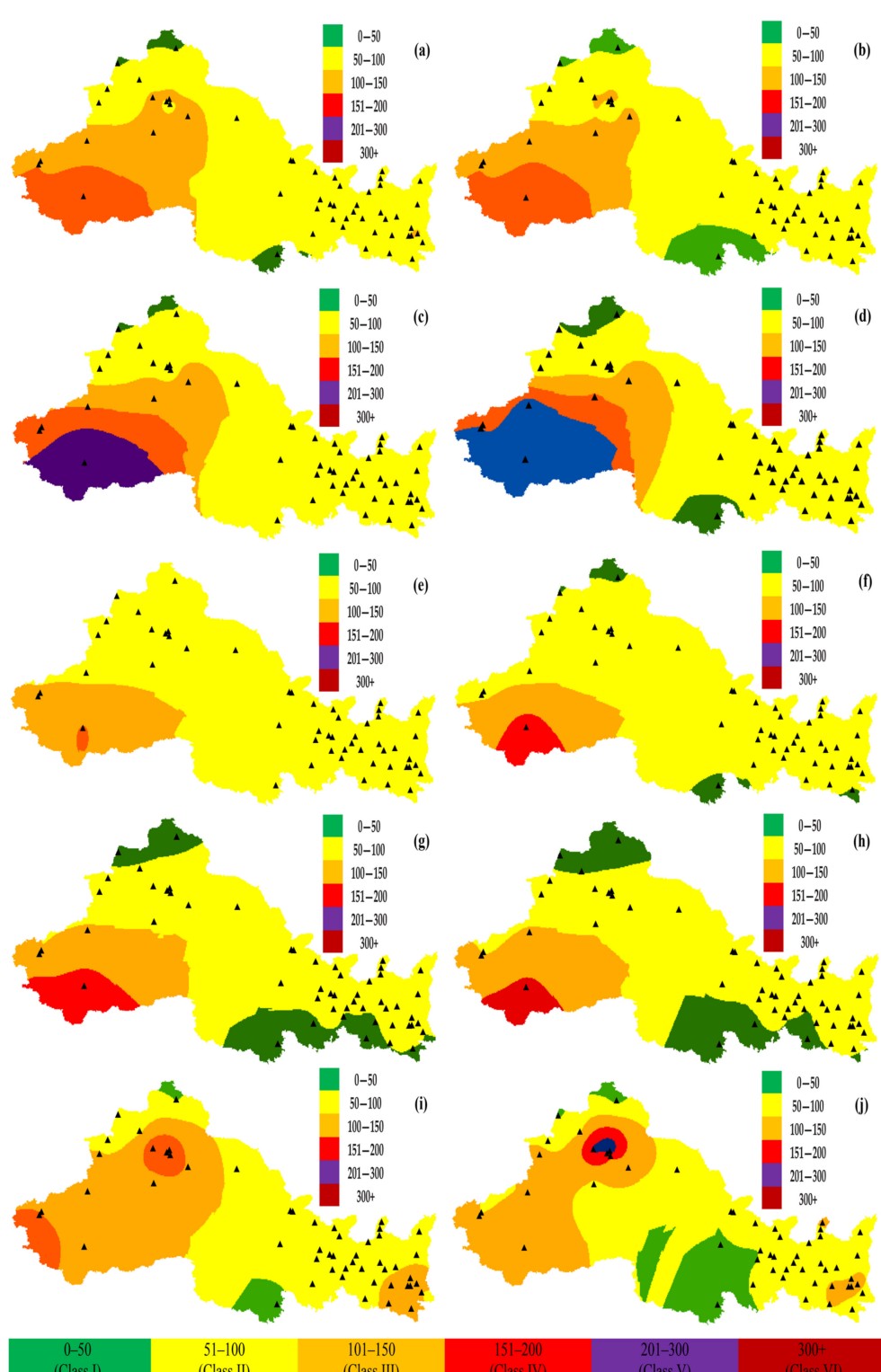

**Figure 6.** Annual (**a**,**b**) and seasonal (spring (**c**,**d**), summer (**e**,**f**), autumn (**g**,**h**), winter (**i**,**j**)) variation of AQI in northwest China (NWC) during 2019 and 2020. Colors represent the different classes of air quality index e.g., green (0–50, good, Class I), yellow (51–100, moderate, Class II), orange (101–150, unhealthy for the sensitive group, Class III), red (151–200, unhealthy for all, Class IV), purple (201–300, very unhealthy, Class V), and maroon (300+, hazardous, Class VI).

### 3.4. Proportion of Six AQI Classes

Figure 7 explains the annual and seasonal (spring, summer, autumn, and winter) proportion of different AQI classes in NWC during 2019 and 2020. In 2020, the proportion of AQI "Class I", and "Class II" improved by 32.14%, and 4%, respectively, while they decreased by 9.13%, 21.35%, and 18.41% for "Class III", "Class IV", and "Class V", respectively, in NWC. In the seasonal variation, the proportion of Class I increased by 22.42%, 50.13%, and 41.95% in spring, summer, and winter, respectively, in NWC during 2020. In the case of monthly variation, the combined proportion of Class I & II was higher in summer (June, July, August), indicating better air quality compared with other seasons (Figure S1).

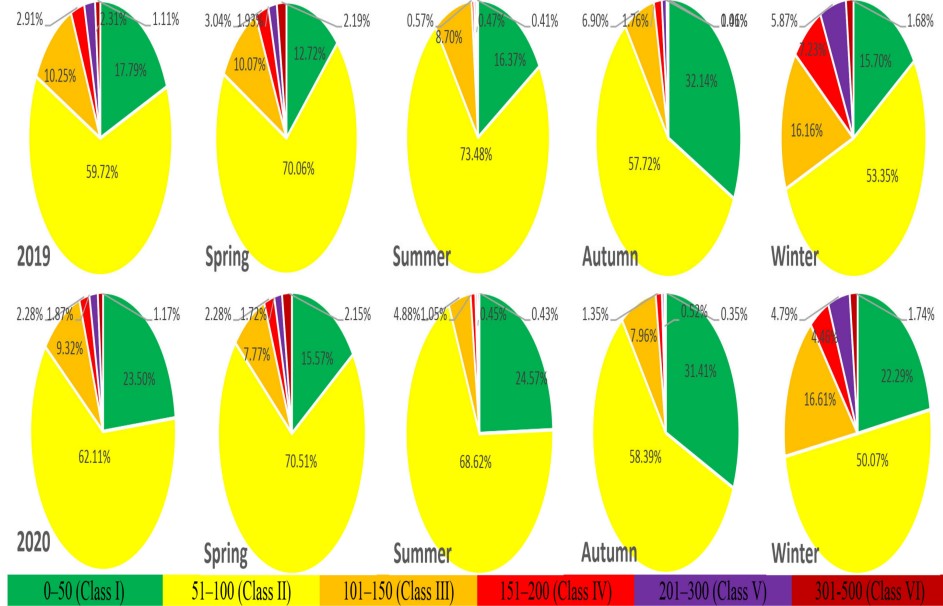

**Figure 7.** Annual and seasonal (spring, summer, autumn, winter) proportion of different air quality index (AQI) classes e.g., Class I (0—50, good, green), Class II (51—100, moderate, yellow), Class III (101—150, unhealthy for the sensitive group, orange), Class IV (151—200, unhealthy for all, red), Class V (201—300, very unhealthy, purple), and Class VI (300+, hazardous, maroon) in northwest China (NWC) during 2019 and 2020.

### 3.5. The Major Pollutants

During the study period, $O_3$ was a major pollutant accounting for 28.06% days followed by $PM_{2.5}$ (23.41%), $PM_{10}$ (21.79%), $NO_2$ (2.28%), $SO_2$ (0.21%), and CO (0.04%). In 2020, the number of days with $PM_{2.5}$, $SO_2$, and $NO_2$ as major pollutants decreased by 46.02%, 51.79%, and 34.77%, respectively, while they increased by 119.45%, 300%, and 3.10% for $PM_{2.5}$, CO, and $NO_2$, respectively (Figure S2).

In the case of seasonal variation, $PM_{10}$ was a major pollutant in spring (43.57%) and autumn (39.32%), $PM_{2.5}$ was a major pollutant in winter (46.98%), and $O_3$ was a major pollutant (67.31%) in summer. The number of days with $PM_{2.5}$ as a major pollutant decreased by 37.5% in spring, while they increased by 33.33%, 47.15%, and 17.65% in summer, autumn, and winter 2020, respectively. Similarly, the number of days with $PM_{10}$ as major pollutant increased by 0.61%, 21.33%, and 8.75% in spring, summer, and autumn 2020, respectively, while they decreased by 32.31% in winter 2020 as compared with 2019. The number of days with $SO_2$, $NO_2$ and $O_3$ as major pollutant decreased in all seasons while increasing slightly for O3 in spring 2020 compared with 2019. The number of days with $O_3$ as a major pollutant was higher in the hotter months (April–September), and $PM_{2.5}$ was higher in colder months (November–February). $PM_{10}$ experienced a "U" shaped curve

with higher concentrations in winter, spring, and autumn, and lower concentrations in summer (Figure S2).

### 3.6. Pollution Days

Any day with one or more pollutants exceeding CAAQS (Grade II) standards is considered as a non-attainment/pollution day. Figure S3 explains the number of pollution days in NWC (SN, XJ, GS, NX, and QH) during 2019 and 2020. In 2020, the number of pollution days in NWC (SN, XJ, GS, NX, and QH) was reduced by 13.59% (22.33%, 13.96%, 13.12%, 27.48%, and 20.53%) as compared with 2019. Similarly, the number of pollution days decreased by 15.41%, 26.79%, 9.66%, and 11.55% in spring, summer, autumn, and winter 2020, respectively, as compared with 2019 (Figure S3).

### 3.7. Statistical Analysis

$PM_{2.5}$ was strongly correlated with $PM_{10}$, CO, $SO_2$, and $NO_2$, indicating emissions from fossil fuel combustion, power plants, vehicular exhaust, industrial emissions, dust storms etc. $O_3$ was negatively correlated with other pollutants in northwest China. The seasonal variation in the correlation between different pollutants was evident. Throughout the study period, all the pollutants were positively correlated with each other except $O_3$. A weak positive correlation occurred between $O_3$ and other pollutants in summer (Table S2).

## 4. Discussion

Based on the results above, we concluded that the pollution level improved significantly in NWC during the COVID-19 outbreak (2020) due to strict epidemic prevention and control measures. In 2020, $PM_{2.5}$ improved significantly in SN and QH, while it improved slightly in GS and XJ due to strict epidemic prevention measures e.g., industrial closure, traffic stagnation, etc. that significantly helped in pollution reduction [27,29,30]. Even after the epidemic prevention measures, NX experienced an increase in pollution, GS and XJ experienced minor improvements that indicate the influence of increased coal consumption, civil heating, industrial activity, etc. [6,13,19,27,44–48]. $PM_{2.5}$ mainly originates from industrial activities, coal consumption, power generation, biomass burning, automobile exhausts, construction activities, road dust, etc. [7,8,10,19,45,49–51]. In 2020, 64.15% of cities in NWC experienced an improvement in $PM_{2.5}$ with the highest number of cities in SN (90%) followed by QH (87.5%), GS (57.15%), XJ (53.25%), and NX (20%), respectively, because the lockdown period was not consistent throughout NWC, and experienced significant spatial and temporal variation. With $PM_{10}$, all regions e.g., SN, QH, XJ, NX, and GS experienced pollution reduction due to strict epidemic prevention measures [27,29,30]. $PM_{10}$ mainly originates from natural sources e.g., sand storms, haze events, etc. as well as from anthropogenic sources e.g., developmental activities, industrial emissions, traffic emissions, road dust etc. [11,13,19] that were largely reduced during lockdowns, and helped significantly in pollution reduction [27,30,35,38]. XJ's southern region experienced the highest $PM_{10}$ pollution level due to increased emissions from natural sources, e.g., Taklimakan deserts [52–55]. About 73.58% of cities in NWC experienced a decrease in $PM_{10}$ with the highest number of cities in QH (90%), followed by XJ (75%), GS (42.85%), and NX (20%) in 2020. Multiple studies experienced particulate reduction during the COVID-19 outbreak, e.g., [27] observed that $PM_{2.5}$ and $PM_{10}$ reduced by 30.1% and 40.5% in Hubei province [27], 46.5% and 48.9% in Anhui province [30], and 5.93% and 13.66% in 44 cities of north China [31].

In the case of gaseous pollutants, $SO_2$, $NO_2$, CO, and $O_3$ decreased in all provinces and decreased in 71.7%, 88.68%, 83.02%, and 71.7% in cities of NWC, respectively, during the COVID-19 year of 2020. Industrial emissions, coal burning, fossil fuel burning, power generation, traffic exhausts, etc., are major sources of $SO_2$, $NO_2$, and CO [45,56–60]. Due to epidemic prevention and control measures, industrial buildings were closed, energy demand was decreased, traffic was stagnant due to "stay at home", "self-quarantine", and

"social distancing" policies that significantly reduced pollution in northwest China during 2020. Multiple studies e.g., [27] observed that $SO_2$ decreased by 33.4% in Hubei, 52.5% in Anhui [30], 6.76% in north China [31], 30% in Wuhan (origin of the viral outbreak) [29], 20 to 30% reduction in China, Spain, France, Italy and northwestern parts of the USA. Similarly, $NO_2$ and CO decreased by 61.4% and 27.9% in Hubei province [27], 52.8%, 36.2% in Anhui province [30], 24.67%, and 4.58% in north China [31]. Similarly, [38] observed that $NO_2$ and CO decreased by 30% in east China, and 20% in Wuhan, respectively, during the lockdown. $O_3$ is a secondary pollutant formed due to a photochemical reaction between VOCs and NOx [27,30,61]. In contrast to other areas [4,27,30,37,62], the average concentration of $O_3$ decreased by 2.15% in NWC, mainly because of a reduction in $O_3$ precursors during lockdown period.

Generally, $PM_{2.5}$, $PM_{10}$ (except XJ in 2020, GS (2019, 2020), NX (2019), QH (2020)), $SO_2$, $NO_2$, and CO observed the same seasonal variation, e.g., highest in winter and lowest in summer. Higher pollution in winter is associated with increased coal combustion, civil heating, power generation, fossil fuel burning, industrial activity, vehicular exhausts, and stagnant meteorology [13,44,46,51,63]. The exception was that, the concentration of $PM_{10}$ was higher in XJ during spring 2020 than in 2019, which indicates the ongoing influence of natural sources, e.g., sand storms, deserts, mineral dust, etc. [52–55]. In contrast to other pollutants, the concentration of $O_3$ was higher in summer than winter [13,46,51,64] due to lower NOx levels in winter, as NOx levels decrease the $O_3$ depletion and enhance the accumulation of $O_3$, as well as higher temperatures favor ozone production [23,27,30,36,37, 65–70]. Lockdown caused minor changes in seasonal variation instead of major changes because the lockdown pattern was not regular and largely varied spatially and temporally.

$PM_{2.5}/PM_{10}$ reflect air quality, pollution sources and origin e.g., a higher $PM_{2.5}/PM_{10}$ ratio indicates an increased proportion of $PM_{2.5}$ mainly emitted from anthropogenic activities [52,53,55]. In 2020, the $PM_{2.5}/PM_{10}$ ratio increased by 2.56%, and 60.38% of cities experienced an increase in the $PM_{2.5}/PM_{10}$ ratio in NWC. This increase is associated with a minor reduction in $PM_{2.5}$ as compared with $PM_{10}$ in 2020. As the share of $PM_{2.5}$ increases, the $PM_{2.5}/PM_{10}$ ratio also increases [13]. In general, the $PM_{2.5}/PM_{10}$ ratio was higher in winter (low temperature), compared to summer (high temperature) due to increased $PM_{2.5}$ emissions from coal combustion, civil heating, biomass burning, industrial emissions, and stable atmospheric conditions helped with stagnation and accumulation of pollution [13,71–73]. In 2020, the $PM_{2.5}/PM_{10}$ ratio decreased in spring and summer, while it increased in autumn and winter due to an increase in $PM_{2.5}$ and minor reduction of $PM_{2.5}$ compared with $PM_{10}$.

In 2020, the AQI improved by 4.70% (10.26%, 2.25%, 2.73%, 0.31%, and 9.74%) in NWC (SN, XJ, GS, NX, and QH), and 77.36% (41/53) of cities in NWC experienced AQI improvement. This improvement is associated with a reduction in criteria pollutants during lockdown [27–39]. The AQI improvement throughout NWC was not uniform because of irregular lockdown periods, air quality deteriorated in some areas, and improved in some areas. In general, smaller cities (<1 million) experienced greater improvement with some exceptions e.g., Shizuishan (0.73 million) and Linxia (0.25 million) experienced deterioration by 3.23% and 12.79%, respectively. Similarly, some big cities (>5 million), e.g., Xian, Weinan, Xianyang, experienced AQI improvement of more than 12%. In seasonal variations, the highest AQI occurred in winter because the concentration of criteria pollutants was higher in winter due to increased anthropogenic emissions in winter and stable atmospheric conditions [11,19,71–74]. In 2020, the AQI improved in all seasons, e.g., spring (83.02%), summer (13.21%), autumn (52.83%), and winter (62.26%) compared with 2019. In NWC, the combined proportion of AQI "Class I", and "Class II" improved by 10.46% in 2020 compared with 2019, which indicates improvement in air quality [13,71]. In 2020, the highest combined proportion of AQI "Class I", and "Class II" occurred in summer followed by autumn, spring and winter, increased in all seasons except autumn as compared with 2019. The combined proportion of AQI "Class I" and "Class II" improved in all provinces except in winter for NX and autumn for SN, GS, NX, and QH in 2020 compared with 2019. Similar studies, e.g., [27] observed that Class I, and Class II increased significantly, while

the proportion of Class III, Class IV, Class V, and Class VI decreased in Hubei and Anhui province during lockdown [30].

In 2020, the number of days with $PM_{10}$ and $O_3$ as major pollutants increased, while the number of days with $PM_{2.5}$, $SO_2$, and $NO_2$ as major pollutants decreased in NWC concerning 2019. Due to limited anthropogenic activity e.g., industrial closures, traffic stagnation, movement restrictions etc., the concentration of $PM_{2.5}$, $SO_2$, and $NO_2$ decreased significantly [23,27,30,31,36,38]. $PM_{2.5}$ was a major pollutant in winter, indicating anthropogenic emissions, e.g., civil heating, industrial emissions, etc. [13,47,48,71]. $PM_{10}$ was a major pollutant in spring and autumn, while $O_3$ was a major pollutant in summer as higher temperatures favor ozone accumulation [35,64–69]. In the southern part of XJ (Kashgar), the number of days with $PM_{10}$ as a major pollutant was higher due to emissions from natural sources, e.g., Taklimakan desert, sand storms [52–55,75]. Any day with one or more pollutants exceeding CAAQS (Grade II) standards is considered as a non-attainment/pollution day [27,30]. During 2020, the number of pollution days decreased by 13.59% in NWC, which indicates improved air quality associated with a reduction in anthropogenic activities. Similarly, the number of pollution days decreased by 15.41%, 26.79%, 9.66%, and 11.55% in spring, summer, autumn, and winter, respectively, which indicates that the ambient air quality was improved significantly in 2020. A strong correlation between all the criteria pollutants indicates mutual emission sources. $PM_{2.5}$, mainly originates from anthropogenic activity e.g., fossil fuels, developmental activity, industrial activity, etc. [7,8,11,15]. Such activities also contribute to SO, $NO_2$, CO and $PM_{10}$. Multiple studies have concluded that coal burning is a major source of $PM_{2.5}$ [8,76] and a major source of $SO_2$; vehicular emissions are a major source of $PM_{2.5}$ [49,77,78] and release $NO_2$ and CO as well. Similarly, road dust mainly consists of $PM_{10}$, which contributes a significant share to $PM_{2.5}$ [10,18,49,77,79].

In short, ongoing rapid economic development, industrialization, urbanization, motorization, natural events, and adverse meteorology have deteriorated ambient air quality in NWC [2,9,16,18,80–83]. The SARS-CoV-2 proved to be a blessing in disguise as the associated lockdowns put in place to prevent the spread of the viral outbreak resulted in a significant reduction in $PM_{2.5}$, $PM_{10}$, $SO_2$, $NO_2$, CO, $O_3$, AQI, and the number of pollution days in NWC decreased during 2020 compared to 2019.

## 5. Conclusions

This study collected hourly monitoring data of ambient air pollutants from 53 cities located in five provinces of northwest China (NWC) from January 2019 to December 2020 to show the collective effect SARS-CoV-2 had on ambient air quality in 2020 as compared with 2019. The results showed that the average concentrations of $PM_{2.5}$, $PM_{10}$, $SO_2$, $NO_2$, CO, and $O_3$ improved by 2.72%, 5.31%, 7.93%, 8.40%, 8.47%, and 2.15% in NWC, respectively, during 2020. The annual average concentration of $PM_{2.5}$ failed to comply in SN, XJ, and NW; $PM_{10}$ failed to comply in SN, XJ, NX, and NW, while $SO_2$ and $NO_2$ complied with CAAQS Grade II standards (35 $\mu g/m^3$, 70 $\mu g/m^3$, 60 $\mu g/m^3$ and 40 $\mu g/m^3$, annual mean) in SN, XJ, GS, NX, QH, and NWC. All the pollutants experienced their highest pollution level in winter except ozone, with varying degrees of spatial distribution. The AQI improved by 4.67% (higher in cities with low population, with some exceptions) in NWC and experienced the highest improvement in SN followed by QH, GS, XJ, and NX. The AQI improved in all seasons with the highest increase in summer, followed by winter, spring and autumn. Significant improvements in the AQI occurred in winter (December to February) and spring (March to April) when lockdowns, industrial closures etc. were at their peak. In NWC, $O_3$ was a major pollutant followed by $PM_{2.5}$, $PM_{10}$, $NO_2$, $SO_2$, and, CO with different spatial and temporal variations, e.g., $PM_{2.5}$ in winter, $PM_{10}$ in autumn and spring, and $O_3$ in summer. A strong correlation occurred between all pollutants except $O_3$. This paper comprehensively discussed the impact of SARS-CoV-2, and associated lockdowns on air pollution in NWC and calls for future detailed assessment focusing on health risk assessment and the impact of meteorology, etc.

**Supplementary Materials:** The following are available online at https://www.mdpi.com/article/10.3390/atmos12040518/s1, Figure S1: Monthly variation in proportion of the air quality index (AQI) classes e.g., Class I (0–50, good, green), Class II (51–100, moderate, yellow), Class III (101–150, unhealthy for sensitive group, orange), Class IV (151–200, unhealthy for all, red), Class V (201–300, very unhealthy, purple), and Class VI (300+, hazardous, maroon) in northwest China (NWC) during 2019 and 2020; Figure S2: Annual, seasonal (a) and monthly (b) proportion of various major pollutants in northwest China (NWC) during 2019 and 2020. Descriptions are as follows: green (PM2.5), yellow (PM10), gray (SO2), orange (NO2), blue (CO), and light green (O3). The abbreviations are as follows: PM2.5 (fine particulate matter), PM10 (coarse particulate matter), SO2. (Sulfur dioxide), NO2 (nitrogen dioxide), CO (carbon monoxide), and O3 (ozone); Figure S3: Annual and seasonal proportion of pollution days in northwest China (NWC) during 2019 and 2020. Descriptions are as follows: blue bar (spring), orange bar (summer), gray bar (autumn), yellow bar (winter), and blue line (annual mean). Table S1: Annual average concentration of Pm2.5, PM10, SO2, NO2, CO, O3, and PM2.5/PM10 in northwest China (NWC) during 2019 and 2020. Table S2: Annual and seasonal correlation between different pollutants in 2019 and 2020.

**Author Contributions:** Conceptualization, S.Z.; data curation, S.Z.; investigation and data validation, J.L., M.Z.S. and I.S.; methodology, S.Z.; software, S.Z.; writing—original draft preparation, S.Z.; review and editing, J.L., M.Z.S., S.A., S.Z., I.S; supervision, J.L.; funding acquisition, J.L. All authors have read and agreed to the published version of the manuscript.

**Funding:** The National Natural Science Foundation of China (No. 21667026) and the Social Science Foundation of Xinjiang Production and Construction Corps (No. 18YB13) funded this work.

**Institutional Review Board Statement:** Not applicable.

**Informed Consent Statement:** Not applicable.

**Data Availability Statement:** The data presented in this study are available upon request from the corresponding author.

**Acknowledgments:** I would like to acknowledge Jianjiang Lu, research scholar Anam Arshad, and Babar Amin from Shihezi University, China for helping and guiding throughout the preparation of this paper.

**Conflicts of Interest:** The authors declare no conflict of interest.

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
