# Peer review of "Impact of SARS-CoV-2 on Ambient Air Quality in Northwest China (NWC)"

_atmosphere, doi:10.3390/atmos12040518_

Round 1
Reviewer 1 Report
- How the study is adding knowledge to already published works, where it was shown that during the lockdown period, the concentration of pollutants are decreasing over China?
- Abstract is not mentioning anything about seasonality. It projects the work to be focused on lockdown period, but the results are discussing different seasons of 2019 and 2020.
- The objective is not very clear in the present form. It seems that the results are already known as there were a lot of publications on similar topic and over China too.
- Figure 2: Why authors are starting the y-axis from 0 values? As they are trying to project the differences during covid-19 lockdown, the –axis can be modified in each pollutant to bring the differences clearly.
- Figure 3: Difficult to compare one to one. Shouldn’t authors make the plot in a way where 2019 and 2020 are kept side by side for same pollutant to see the difference? IF we go by the results, Figure is indicating that 2020 is having uniformly higher values of moderate NO2 compare to 2019, where there were unhealthy and good both groups. What is the reason of this hike in NO2 values? Is it because of interpolation applied to results?
- Figure 4: legends are to be specified in top panels of figure. Instead of (a) or (b), the legends are in subplots (c) and (e)?
- Discussion and conclusion: What are the salient features observed by the authors? Too long by mentioning all the values again from previous section. Point wise conclusion may be better.
- Please add a table where the authors should compare the findings of their work with earlier published works on COVID-19 and pollution over China. It is required to bring out the additional information/knowledge added by the present work, if any.
- There are many grammatical issues with the manuscript. I suggest authors to go through the text carefully to improve the language scientifically.
Reviewer 2 Report
The manuscript deals with an interesting topic on the influence of low anthropogenic activities due to COVID-19 and lockdowns on air quality conditions.I find it very informative, the paper has many useful and essential information for both the scientific and domestic community and therefore presented manuscript merits publication in the journal Atmosphere. However, I find some lack of details, inconsistent use of abbreviations, region names, pollutants. It was sometimes very hard to follow all of those percentage numbers, I strongly suggest authors to put all of them in tables.
I'm missing some basic information on when were all of those lockdowns and what were AQ conditions in those days. It would be nice to see a comparison of 2019 and 2020 only in those days, and in the end, after how many days of lockdown the AQI was improved?
Minor comments:
Ln 19 - missing comma after PM2.5
Ln 36, 37 – missing something in a sentence, e.g. “was” after “COVID-19” Ln 99 – what is the type of stations that were used in the analysis (rural, urban, ?)
Ln 104 – 24 hourly – daily or hourly averaged?
Ln 139 - is this really sufficient? The correlation coefficient is a measure of linearity, using only this parameter it really doesn't provide clear picture. It is possible that one can have high corr coeff and also high bias. It is mandatory to use a couple of other statistical measures in analysis in combination with corr coef, e.g. bias, mae, mse, rmse, nmse.
Ln 143 - "concertation" -> "concentration"
Ln 143 - 146. Add more information, what kind of average? Annual of seasonal? And how is it spatially averaged? Did the authors apply the Kriging method before, or this is just a value representing an average of measurements across NWC?
Ln 142 - 157. It's a little bit messy here. It is hard to track all of these numbers as there are many pollutants. I suggest putting all of these numbers in Table. Please elaborate more on Figure 2 - what regions, what pollutants etc.
Ln 157. Can you comment on how reliable was applied Kriging method? Are values on Figure 3 corresponding to one on Figure 2?
Ln 180. Why is ration different in seasons? Due to sources? What is the conclusion, why did PM ratio increased in 2020?
Ln 199. Why italic font?
Ln 200. Some percentages are italic?
Ln 206. The figure cannot explain something, however, the authors can use it for explaining. Please correct this sentence. I strongly suggest rewriting this paragraph. The figure is not showing the distribution of classes, please see the comment for Figure 7.
Ln 217. Primary pollutants are pollutants emitted from the source. It is confusing what exactly is primary pollutant here? Maybe major pollutant?
Ln 220. Not clear, the proportion of days with average values or?
Ln 236. CAAQS - use rather an abbreviation as previously in the text
Ln 238. China-NWC - same as the previous comment, use abbreviation NWC as previously in the text
Ln 246 - 248. Can you comment please more on this? What is the source of this information, why correlation of PM2.5 and PM10 indicates emission from fossil fuels?
Ln 257. Use either italic or bold fonts for "Fig" in the text
Ln 270. Please be more consistent in the text. Use full names or abbreviations for regions.
Ln 283. Why using italic font?
Ln 292. Wuhan is the epicenter? I suppose that the authors are referring to Wuhan as the origin of pandemia. Epicenter is the point on the earth's surface vertically above the hypocenter. Please correct this in the text.
Ln 297, 299. Please be consistent in the text, either use "ozone" or "ozone (O3)"
Ln 305. Can authors comment on how come that lockdown didn't cause any difference in seasonal variation between 2019 and 2020. Maybe I'm wrong, when exactly was lockdown in NWC, was it the same period in all of those cities?
Ln 314. Can authors write what exact number of cities is this? Did AQI improved in all cities in the same way, for example, are all of these cities/regions with the same population? Same lockdown periods?
Ln 338- 339. Please replace words: "tells" and "manmade"; Ratio is not "telling", and "mandmade" should be "anthropogenic"
Ln 359. Need more information in Conclusion. Did AQI improved in the seasons when lockdown was present, or it was during all of seasons no matter of lockdown? What regions had the highest decrease.
Fig 2. Very confusing figure. I suggest making one legend for all plots (as on some plots the half of legend is overlapped with another plot, e.g. g) ), add more space between plots, add title on plot (what pollutants is analyzed), etc.
Fig 3. The map is nice, I suggest adding dots with city locations, or at least county boundaries.
Fig S1: AQ or AQI? Please add colorbar aside figure.
Fig 4. Same comment as for Fig 2. Check Figure caption - 2015 - 2018?
Fig 5. Same comment as for Fig 3. What are the a-j for?
Fig 6. Same comment as for Fig 5.
Fig 5&6 It would be useful also to plot the difference between a and f, b and g, etc (in percentages).
Fig 7. The caption is unclear? How is this distribution? It is a pie plot. Distribution can be spatial or temporal. Please elaborate more in the caption of what is exactly shown on figures.
All Figure caption need more information. On maps - please add that Kriging was applied, on plots - what kind of measurements is behind (average?). Every figure needs to be standalone.
Reviewer 3 Report
- Air quality could be improved by reducing anthropogenic emissions due to less activities during lockdown period. It would be better to describe the variation or decline of human activities in different provinces during lockdown.
- There are totally 53 cities in 5 provinces. The data are presented as ensemble mean±S.D. The data of PM10 and NO2 indicated that SD > mean. It would be better to analyzed air quality by grouping data base in each province. It would show the difference clearly.
- What are the differences of air quality between 2019-2020 for those cities at higher conc. level and at lower conc. Level, respectively? It would be interested if the authors could present the impact (conc. increasing or decreasing) in different groups of cities.
- Are there any differences between 2019-2020 for various air pollutants in large cities and small cities? Air pollutant concentrations declined, except ozone. (Fig.4) Please check the characteristics of the regions with increasing ozone conc.
- The conclusions do no clearly present the differences of each pollutant caused by lockdown. Air quality level, seasonal effect and variations among various provinces, and scale of city are also important information to understand the impacts caused by pandemic lockdown.

Round 2
Reviewer 1 Report
I am happy to see authors improvements made in the manuscript. Please keep up the good work.
Reviewer 2 Report
I thank the authors for replying to all of my comments. I hope that the authors are satisfied with the achieved result.
Be careful, some figures were stretched in this newer version of the manuscript.
Cheers.